# Occupational Precariousness of Nursing Staff in Catalonia’s Public and Private Nursing Homes

**DOI:** 10.3390/ijerph16244921

**Published:** 2019-12-05

**Authors:** Ana Mari Fité-Serra, Montserrat Gea-Sánchez, Álvaro Alconada-Romero, José Tomás Mateos, Joan Blanco-Blanco, Eva Barallat-Gimeno, Judith Roca-Llobet, Carles Muntaner

**Affiliations:** 1Department of Nursing and Physiotherapy, University of Lleida, Lleida 25198, Spain; afite@infermeria.udl.cat (A.M.F.-S.); joan.blanco@infermeria.udl.cat (J.B.-B.); ebarallat@dif.udl.cat (E.B.-G.); judithrl@dif.udl.cat (J.R.-L.); 2Group for the Study of Society Health Education and Culture, GESEC, University of Lleida, Lleida 25198, Spain; alvaro.alconada@dif.udl.cat; 3Health Care Research Group, GRECS, Biomedical Research Institute of Lleida, Lleida 25198, Spain; jt.mateosgarcia@gmail.com; 4Public Health Research Group, University of Alicante, Alicante 03690, Spain; 5Faculty of Nursing, Dalla Lana School of Public Health, University of Toronto, Toronto, ON M5T 3M7, Canada; carles.muntaner@utoronto.ca

**Keywords:** nurses, nursing homes, work conditions, employment conditions

## Abstract

Nursing staff who provide care in the nursing homes of Catalonia have more precarious work conditions, including more demanding schedules and work overload, than those in other areas of care. This situation entails two major problems: Detrimental health results for nurses who face psychosocial and physical risks and a negative impact on the care provided to patients, with a decrease in the quality of care. This study aimed to describe the precarious employment situation of nursing staff in nursing homes. We carried out a descriptive study based on the employment precariousness scale (EPRES), which was administered to a sample of 239 nurses and nursing assistants working in public and private nursing homes in Catalonia. The highest level of job insecurity occurred among nursing assistants and in privately managed nursing homes. The precariousness of the working conditions of nursing staff poses a risk both to the workers themselves and to the people they tend to. For this reason, there is a need for greater knowledge on the scale of the problem and the implementation of appropriate legislative measures to alleviate it.

## 1. Introduction

The precariousness of paid work is defined as insecurity in labor relations and the increase in temporary employment [1]. This precariousness is associated with physical and mental health risks [2,3]. These working conditions have spread in recent decades in most Western countries and labor sectors [4]. It is the youngest workers and women who endure the most precarious conditions [5,6]. Although with some variations between regions and occupations, health care is one of the occupations that most suffers this trend in the labor market [7]. As a result of the lack of funding due to austerity policies, understaffing and workload intensification are occurring in the health care services [8]. Furthermore, the gender gap effect in the labor market is also a challenge, as more than two-thirds of the health care workers worldwide are women [7].

The degradation of the working conditions of health care workers is especially widespread in nursing homes [9]. Due to the ageing of the population, the number of dependent patients is increasing, leading to the need for services specialized in their care [10]. At the same time, the cognitive, functional, or nutritional deterioration associated with these types of patients requires complex care [11]. This situation results in less retention of staff and more problems of burnout and stress than in other health care services [12]. Furthermore, nursing staff in nursing homes are at a higher risk of developing musculoskeletal problems [13]. Previous literature has identified specific stressors like overwork, understaffing, and low income [14].

In Spain, 6.5% of the total number of employees are social and health care workers, most of them in elderly care [7]. As in other surrounding countries, nursing homes are mostly managed by private companies (71% in 2017), the working conditions of which are committed to the profit margin [15,16]. Nursing professionals in Spain are among the sectors in which job insecurity has increased in recent years, accentuated especially with the Great Recession [17]. Understaffing due to the lack of hiring new personnel has made longer working hours necessary, with fewer breaks and more staff rotations to provide adequate coverage [18]. In addition, the low remuneration of some activities forces nurses and nursing assistants to combine more than one job [19].

About 83% of nursing homes in Catalonia are managed by private companies [16]. Although it is difficult to find local data about the situation in nursing homes in Catalonia, we can extract some descriptive information from a previous study about the health, lifestyles, and work conditions of 2258 nurses in Catalonia. From this sample, 5.3% (*n* = 120) worked in nursing homes and 28% of them had longer working hours. The nursing homes staff from this study showed major overwork and psychosocial factors than in other health services [20].

This precariousness in the working conditions of nursing staff has an influence not only on the health of the workers, but also on the quality of care provided and the well-being of patients [21,22]. In addition, the psycho-emotional burdens of this type of work for nurses have a significant impact on the consequences for their mental health [23]. There is little scientific literature on the presence of precarious employment in nursing homes. Bearing in mind the indications that these workers suffer from poor working conditions, a greater empirical contribution is required in order to know their causes and consequences. This phenomenon could lead to two major problems: On the one hand, detrimental health results for nurses and, on the other hand, a negative impact on the care provided to patients. This is why a detailed analysis of the situation is necessary to improve current policies. The aim of this work is to describe the precarious working situation of nurses and nursing assistants in public and private nursing homes in Catalonia.

## 2. Materials and Methods 

We carried out a cross-sectional observational study in 19 nursing homes, including publicly and privately managed homes, in Catalonia. We selected the nursing homes based on characteristics according to the review of previous literature and the aim of our study: (1) geographical criteria: Rural or urban area and (2) type of nursing home: Public, private, or public with private management and mixed management systems, covering the whole region of Catalonia. In each nursing home, we carried out a nonprobability convenience sampling of nurses and nursing assistants who worked in these nursing homes, based on the criteria of accessibility and availability [24]. The total sample consisted of 239 participants, of whom 53 were nurses and 186 were nursing assistants. Both groups were selected according to their accessibility in the visits during the three shifts (morning, afternoon, and night). Although the size of the groups was different, both were representative of the total of the two occupational groups. The response rate of nurses was 19.1% of the total of 278 who worked in nursing homes, while that of nursing assistants was 18.7% of the total of 994 who worked in nursing homes. 

The measuring instrument used was the employment precariousness scale (EPRES), validated in the Catalan context and widely used in health research [25]. It is a scale to know the existence of labor precariousness and its casual factors. The EPRES scale fits the standards of reliability and validity, with good internal consistency, as required for this study. EPRES was validated in Spain in 2004, for general salaried workers. In 2010, the version was revised in order to overcome the limitations identified in the 2004 validation, so that an updated version was obtained [26]. This scale includes six dimensions: Temporality, disempowerment, vulnerability, wages, rights, and the exercise of rights. This is a five-item Likert psychometric scale to determine the levels of agreement and frequency. 

In addition to the overall result of the EPRES questionnaire measured from 0 (less precarious) to 4 (more precarious), we collected subscales that measured in the same way (temporality, salary, disempowerment, vulnerability, rights, and exercise of rights). We also collected other variables such as (1) type of occupational group (nurse or nursing assistant); (2) province (Lleida, Barcelona, Girona, or Tarragona); (3) city context (urban, medium, or rural); (4) management systems (public, private, or public—privately managed/mixed); (5) sex (male or female); (6) age; (7) years of experience; and (8) salary.

Each participant’s information was obtained through the questionnaire conducted by previously trained nursing professionals. We performed a descriptive statistical analysis with central trend measurements and a bivariate analysis with the Student’s t-distribution test, establishing statistical significance with a *p*-value < 0.05. We carried out the systematization of the data using Microsoft Excel and the analysis with the statistical analysis software, R-project.

This study complied with the Spanish regulations on research studies whose participants were employees. Additionally, the ethical principles of the Declaration of Helsinki were followed, and the protocol was approved by the Ethics Committee of the Official College of Nurses of Lleida (089352). All content and objectives of the study were explained to all participants. Only those who voluntarily gave their consent to participate in the study were included. The analysis of the responses was anonymized and protected by the research team.

## 3. Results

Concerning the descriptive variables of the composition of the sample, 92.8% (*n* = 218) of the nurses and nursing assistants were women and their average age was 42 years (standard deviation (SD) = 12.1). Of them, 71.12% (*n* = 170) worked in privately owned or managed nursing homes and 87% (*n* = 208) in urban areas. The average salary was 1026 euros (SD = 355) and the mean years of experience was 11.4 years (SD = 7.89).

The mean score of the EPRES questionnaire was 0.83 points (SD = 0.38). Regarding the dimensions of the questionnaire, the dimension with the highest average score (highest precariousness) was salary with 1.6 (SD = 0.68), followed by vulnerability (1.04; SD = 0.87), temporality (0.86; SD = 0.94), disempowerment (0.67; SD = 0.97), exercise of rights (0.66; SD = 0.71), and rights (0.16; SD = 0.29).

Regarding the bivariate analysis, some significant differences in the average score of the EPRES questionnaire and its dimensions were noted (Table 1). Firstly, nursing assistants obtained a mean score of 0.15 points higher than nurses (*p* < 0.01). This difference seems to come from the salary dimension of EPRES, in which the nursing assistants exceeded the nurses by 0.6 points (*p* < 0.01). Another noteworthy result was the difference between the types of nursing homes, with less precariousness declared by the workers of publicly owned or managed nursing homes. This significant difference was maintained in all dimensions except that of rights, which was most notable in the dimension on temporality (Table 1).

Another important finding was the negative correlation between the EPRES questionnaire score and age (−0.25; *p* < 0.01) and experience (−0.44; *p* ≤ 0.01) of the workers; those younger and with less experience had a higher precariousness score, motivated by the temporality dimension (Table 2). We also found a negative relationship between the EPRES score and the salary of the workers (−0.45; *p* ≤ 0.01), which was significant for all dimensions except disempowerment and rights (Table 2). These negative correlations were maintained in the stratified analysis for nurses and nursing assistants.

## 4. Discussion

In this study, we describe the extension and characteristics of precarious employment in nurses and nursing assistants who work in nursing homes in Catalonia. Although, due to the sample size of the study, some conclusions can be cautiously drawn that highlight the importance of the problem and its relevance to public health. 

Firstly, it seems clear that the participant nurses and nursing assistants who provide care in nursing homes endure precarious work conditions. In this sense, previous literature has shown the risks that this situation poses, not only for their health, but also for that of the patients, with worse outcomes and less satisfaction [27,28]. These precarious conditions have some defining characteristics; from our results, we can extract that workers with less age and experience are the ones that suffer more precariousness. This characteristic is in line with the general trend in the labor market, whereby younger people suffer worse working conditions, with greater instability and lower wages [5]. This finding agrees with previous literature that points out that the years spent in nursing favor a major satisfaction and adaptation with the work environment [29,30]. On the other hand, our results also show that nurses and nursing assistants working in privately owned or managed nursing homes also suffer greater precariousness. This finding follows the line of thinking that questions the privatizing trends of these types of services, in which the conditions of the workers and the quality of the attention that they provide are subordinate to the market logic [11,15]. 

Secondly, the analysis of the dimensions of the EPRES questionnaire has provided us with relevant findings in order to understand the causes of this precariousness. The most significant dimension in our results has been salary. This dimension fits with previous research about the effects of working conditions [22], but does not seem to have an effect on job satisfaction, which is modulated by other factors [31,32]. Low wages, as in other labor sectors, entail risks for the mental and physical health of workers due to the lack of financial resources, which can lead these employees to assume worse working conditions or a greater workload [3]. Furthermore, the dimension of temporality has also been significant, which, in the same way, influences the mental and physical health of workers due to the burden of perceived job insecurity and continuous job changes [33]. These findings have highlighted areas in which the interventions to improve working conditions of nursing staff should be directed in nursing homes and health care work [7,11].

Some limitations must be taken into account when interpreting our results. The sample size and its contextualization do not allow inferences to be made at the national or international level; however, it does provide an image of the situation of nursing staff in Catalonian nursing homes. Likewise, the convenience sampling strategy has some limitations, even though, sample selection was designed to represent the workers of nursing homes in Catalonia. Moreover, the EPRES questionnaire is not exclusive for its application to health care workers, and is not specific to nursing; thus, it may not cover all the specific characteristics of this sector, such as the emotional burden to which the workers are exposed. Nevertheless, it does provide an approximation of the working conditions of these workers.

## 5. Conclusions

The knowledge of conditions in which work is carried out in nursing homes is a basic objective, both to ensure the safety of employees and to guarantee the best care for patients. Our findings show that variables, such as the type of management of a nursing home and years of experience or professional rank, influence precariousness, which is manifested mainly in low salaries, temporariness, and the physical and psycho-emotional burdens associated with the type of tasks performed by female workers. 

We suggest continuing this line of research with different types of studies that allow analyzing the factors that determine the labor precariousness of employees of nursing homes, with the purpose of promoting organizational changes that maximize the quality of both patient care and labor conditions.

## Figures and Tables

**Table 1 ijerph-16-04921-t001:** Mean comparison between sample’s main characteristics of employment precariousness scale (EPRES) results and its six dimensions.

	EPRES	Temporal.	Wage	Disemp.	Vulnerab.	Rights	R. Exercise
Global	0.83 (0.38)	0.86 (0.94)	1.6 (0.68)	0.67 (0.97)	1.04 (0.87)	0.16 (0.29)	0.66 (0.71)
Nurse/N. Assistant	0.72 (0.39)/0.87 (0.37) **	0.74 (0.93)/0.9 (0.94)	1.17 (0.66)/1.72 (0.63)	0.7 (0.95)/0.66 (0.97)	0.95 (0.89)/1.07 (0.86)	0.21 (0.3)/0.15 (0.29)	0.58 (0.55)/0.69 (0.75)
Urban/Rural	0.82 (0.36)/0.93 (0.48)	0.85 (0.94)/0.9 (0.93)	1.6 (0.69)/1.59 (0.57)	0.59 (0.9)/1.23 (1.2) **	1.04 (0.87)/1.08 (0.87)	0.16 (0.28)/0.19 (0.35)	0.68 (0.69)/0.54 (0.82) *
Private/Public	0.94 (0.37)/0.59 (0.25) **	1.01 (0.98)/0.49 (0.71) **	1.76 (0.64)/1.21 (0.61) **	0.76 (1)/0.46 (0.85) **	1.17 (0.91)/0.73 (0.68) **	0.15 (0.29)/0.19 (0.28)	0.75 (0.77)/0.44 (0.45) *

Mean (standard deviation (SD)) and comparison significance—* *p* < 0.05, ** *p* < 0.01; Nurse/N. Assistant—punctuation of nurses vs nursing assistants; Urban/Rural—punctuation of staff in urban nursing homes vs in rural nursing homes; Private/Public—punctuation of staff in private nursing homes vs in public nursing homes; EPRES—employment precariousness scale; Temporal.—temporality; Disemp.—disempowerment; Vulnerab.—vulnerability; R. exercise—exercise of rights.

**Table 2 ijerph-16-04921-t002:** Correlation between experience, age, and wage of participants and EPRES results and its six dimensions.

	EPRES	Temporal.	Wage	Disemp.	Vulnerab.	Rights	R. Exercise
Experience	−0.44 **	−0.72 **	−0.13 *	−0.11	−0.03	−0.07	−0.09
Age	−0.25 **	−0.48 **	0.08	−0.06	0.01	−0.08	−0.11
Wage	−0.45 **	−0.35 **	−0.49 **	−0.01	−0.2 **	0.07	−0.15 *

T-value and significance—* *p* < 0.05, ** *p* < 0.01; Experience—experience of participants in years; Age—age of participants in years; Wage—salary of participants in euros (continuous); EPRES—employment precariousness scale; Temporal.—temporality; Disemp.—disempowerment; Vulnerab.—vulnerability; R. exercise—exercise of rights.

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
