# Peer review of "Occupational Precariousness of Nursing Staff in Catalonia’s Public and Private Nursing Homes"

_ijerph, 2019, doi:10.3390/ijerph16244921_

Round 1
Reviewer 1 Report
The authors have developed and described the study well. I have the following minor suggestions:
On p.2, the authors describe their general approach to sampling and their final sampling pool. I'd like to see more detail about the sampling approach. For instance, is the ratio of nurses to nursing assistants reflective of the ratio in the population? Were participants evenly or proportionally drawn from the two criteria? On p.2, the authors explain that "each participant's answers were obtained through an interview..." Does this mean that the participant completed the EPRES questionnaire during the interview or that after completing the EPRES questionnaire, each participant was interviewed and the results were translated into the variables discussed on lines 72-76? On p.4, the authors discuss the limitations based on sample size. They might consider discussing limitations in their theoretical sampling strategy as well.
Reviewer 2 Report
INTRODUCTION
The study problem is not contextualized in a clear way.
I would propose to the authors a contextualization from the real international, national and local problem.
DISCUSSION
All significant results should be commented and justified with updated bibliography, in this article the presence of citations in this section is scarce.
BIBLIOGRAPHY
Updated bibliography (avoid abusing appointments that are> 5 years old (the current one is not counted. 16/28 appointments 60%.
Author Response
November 29, 2019
Dear Reviewer:
We would like to express our gratitude for the interest showed in our Manuscript ID ijerph-653980, entitled "Occupational Precariousness of Nursing Staff in Catalonia’s Public and Private Nursing Homes", as well as, by your constructive insights and recommendations, which have been taken into account in our revision of the manuscript.
Throughout this letter, we describe in detail the way in which we have answered to yours contributions. We have revised the text, highlighting the changes into the manuscript by using colored highlighted text in MS Word.
Similarly, we inform you that the English of the manuscript has been professionally proofread by Editage company (www.editage.com).
***
Reviewer’s Comments to Authors
Comment 1
INTRODUCTION. The study problem is not contextualized in a clear way. I would propose to the authors a contextualization from the real international, national and local problem.
Answer 1
We greatly appreciate your comment. It has certainly led us to think much more about the contextualization of the problem. Hence, we have made changes in the introduction section. We believe that these changes contribute to a greater description of the study problem and better clarification of the context. We have introduced the first paragraph with an international overview of the problem, based on reports of international organizations like International Labour Organization or European Union. As a general introduction to this point you can now read the following:
“The precariousness of paid work is defined as insecurity in labor relations and the increase in temporary employment [1]. This precariousness is associated with physical and mental health risks [2,3]. These working conditions have spread in recent decades in most Western countries and labor sectors [4]. However, it is the youngest workers and women who endure the most precarious conditions [5,6]. Although with some variations between regions and occupations, health care workers are one of the occupations that most suffer this trend in the labor market [7]. As a result of the lack of funding due to austerity policies, understaffing and workload intensification are occurring in the health care services [8]. Furthermore, the gender gap effect in the labor market is also a challenge, as more than two-thirds of these workers worldwide are women [7].
The degradation of the working conditions of health care workers is especially widespread in nursing homes [9]. Due to the ageing of the population, the number of dependent patients is increasing, leading to the need for services specialized in their care [10]. At the same time, the cognitive, functional or nutritional deterioration associated with these types of patients requires complex care [11]. This situation results in less retention of staff and more problems of burnout and stress than in other health care services [12]. Furthermore, nursing staff in nursing homes have more risk of musculoskeletal problems [13]. Previous literature has identified specific stressors like overwork, understaffing and low income [14].”
In order to provide a national and local contextualization of the study problem, we also added a deeper explanation of the Spanish and Catalan context. The new paragraph is as follow:
“In Spain, 6.5% of the total number of employees are social and health care workers, most of them in elderly care [7]. As in other surrounding countries, nursing homes are mostly managed by private companies (71% in 2017), the working conditions of which are committed to the profit margin [15,16]. Nursing professionals in Spain are among the sectors in which job insecurity has increased in recent years, accentuated especially with the Great Recession [17]. Understaffing due to the lack of hiring new personnel has made longer working hours necessary, with fewer breaks and more staff rotating to provide adequate coverage [18]. In addition, the low remuneration of some activities forces nurses and nursing assistants to combine more than one job [19].
Noting the above, 83% of nursing homes in Catalonia are managed by private companies [16]. Although it is difficult to find local data about the situation in nursing homes in Catalonia, we can extract some descriptive information from the previous study about the health, lifestyles and work conditions of 2258 nurses in Catalonia. From this sample, 5.3% (n=120) worked in nursing homes and 28% of them had longer working hours. The nursing homes staff from this study showed major overwork and psychosocial factors than in other health services [20].”
Comment 2
DISCUSSION. All significant results should be commented and justified with updated bibliography, in this article the presence of citations in this section is scarce.
Answer 2
We thank the reviewer for this suggestion and agree that the discussion section could be improved with more citations and deeper justification. Therefore, we have added some comments to our results, supported by a new bibliography. The main paragraph of the discussion section in the new version of the manuscript is as follows:
“Firstly, it seems clear that the participant nurses and nursing assistants who provide care in nursing homes endure precarious work conditions. In this sense, previous literature has shown the risks that this situation poses not only for their health but also for that of the patients, with worse outcomes and less satisfaction [27,28]. These precarious conditions have some defining characteristics; from our results, we can extract that workers with less age and experience are the ones that suffer more precariousness. This characteristic is in line with the general trend in the labor market, whereby younger people suffer worse working conditions, with greater instability and lower wages [5]. This finding agrees with previous literature that points out that the years spent in nursing favor a major satisfaction and adaptation with the work environment [29,30]. On the other hand, our results also show that nurses and nursing assistants working in privately-owned or managed nursing homes also suffer greater precariousness. This finding follows the line of thinking that questions the privatizing trends of these types of services, in which the conditions of the workers and the quality of the attention that they provide are subordinate to the market logic [11,15].
Secondly, the analysis of the dimensions of the EPRES questionnaire has provided us with relevant findings in order to understand the causes of this precariousness. The most significant dimension in our results has been salary. This dimension fits with previous research about the effects of working conditions [22], but does not seem to have an effect on job satisfaction, which is modulated by other factors [31,32]. However, low wages, as in other labor sectors, entail risks for the mental and physical health of workers due to the lack of financial resources, which can lead these employees to assume worse working conditions or a greater workload [3]. Furthermore, the dimension of temporality has also been significant, which, in the same way, influences the mental and physical health of workers due to the burden of perceived job insecurity and continuous job changes [33]. These findings have highlighted areas in which the interventions to improve working conditions of nursing staff should be directed in nursing homes and health care work [7,11].”
Comment 3
BIBLIOGRAPHY. Updated bibliography (avoid abusing appointments that are> 5 years old (the current one is not counted. 16/28 appointments 60%.
Answer 3
We would like to thank the reviewer for this comment. We have updated and expanded the references used in this manuscript. There are 22 new references and only one is from 2014, all others are from 2015 to 2019.
***
Thank you for allowing us to conduct an improvement review following your constructive comments.
On behalf of all authors,
Best regards.
Round 2
Reviewer 2 Report
The changes are correct and conform to what is requested by this reviewer.